# The Process of Identifying Automobile Joint Failures during the Operation Phase: Data Analytics Based on Association Rules

Polina Buyvol *, Irina Makarova, Aleksandr Voroshilov and Alla Krivonogova

Naberezhnye Chelny Institute, Kazan Federal University, Syuyumbike Prosp. 10a,
423812 Naberezhnye Chelny, Russia
* Correspondence: skyeyes@mail.ru

**Abstract:** The increasing complexity of vehicle design, the use of new engine types and fuels, and the increasing intelligence of automobiles are making it increasingly difficult to ensure trouble-free operation. Finding faulty parts quickly and accurately is becoming increasingly difficult, as the diagnostic process requires analyzing a great amount of information. Therefore, we propose an approach based on association rules, a machine learning technique, to simplify the defect detection process. To facilitate its use in a real repair company environment, we have developed a web service that allows a repairman to simultaneously identify nodes with a high probability of failure. We have described the structure and working principles of the developed web service, as well as the procedure for its application, which resulted in the discovery of several useful non-trivial rules. We have presented several rules resulting from the use of this interactive tool, which allow repairers to detect possible defects in the relevant components, during the diagnostic process, quickly and easily. These rules are also well supported and can be used by procurement departments to make tactical decisions when selecting the most promising suppliers and manufacturers. The methodology developed allows the evaluation of the effectiveness of changes in the design and technology for the manufacture and operation of individual vehicle components, analyzing the change in the composition of parts combinations over time.

**Keywords:** association rules; defect analysis; automobile repair; decision support; web service

## 1. Introduction

Today, the growing sophistication of vehicle construction, the introduction of modern engine types, the transition to alternative fuels, and the growing share of electrical and electronic equipment complicates the process ensuring consumers experience trouble-free operation of their vehicles. Diagnostic work includes an increasing proportion of repair and maintenance work and requires a certain level of expertise. Troubleshooting faults during diagnosis quickly and accurately is becoming increasingly difficult, due to the great volume of information to be processed, but it is necessary to restore vehicle performance quickly and accurately. Because a vehicle is a complex technical system consisting of some interacting elements, intelligent analysis methods and instruments are needed to identify complex relationships, determine the most critical components, and establish repair sequences.

At the same time, the likelihood of unmanned vehicles entering public roads around the world is increasing. In this regard, the service concept will change: if for traditional vehicles, on-demand repairs when a malfunction is detected and periodic technical maintenance are accepted, then for unmanned vehicles, daily pre-trip inspections are more appropriate. This change is due to the fact that in the vehicle–driver system for unmanned vehicles the role of the driver is excluded, but indirect signs (smell, sound, noise, vibration) can be used to identify an existing malfunction and predict a possible failure in the near future. Depending on the ability to perform its functions and on compliance with the

requirements established by the normative-technical and design documentation, an object can be in good, fault, and inoperable states. When the limiting state is reached, the object is removed from operation (Figure 1). The complexity of vehicle troubleshooting also lies in the fact that not all malfunctions lead to a fault state in vehicles.

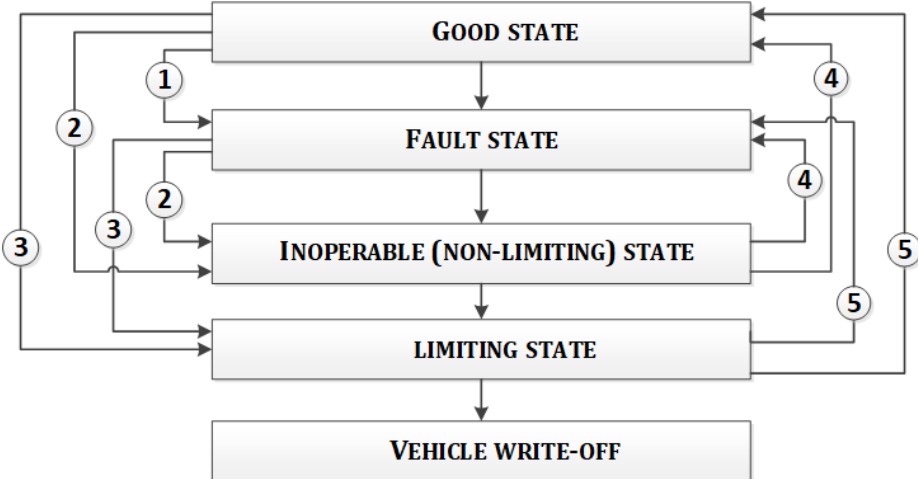

**Figure 1.** Scheme of vehicle states and their corresponding events: 1—damage; 2—failure; 3—marginal failure; 4—restoration; 5—overhaul.

In this case, defective parts affect other joint parts and can lead to a faulty state, resulting in the inoperable state of the entire vehicle. This type of failure is called dependent. This is a failure that occurs as a result of a change in the reliability characteristics of other elements. Thus, being operable at the time of the check, the vehicle may fail suddenly and, as a result, create a potential safety hazard for both the passengers in the vehicle and other road users. These situations can be avoided if all the defective parts are identified in time. Through external manifestation, one can single out explicit failures and implicit (hard to detect) failures, for the identification of which special means are used. In this regard, the urgency of the problem of identifying defective technical systems components increases.

A lot of research on investigating methods to analyze technical systems reliability has been conducted. Typical examples include fault tree analysis [1], logical probabilistic methods [2], failure and effects analysis [3], multivariate failure analysis [4], fuzzy logic [5], failure distribution function construction [6,7], and equipment diagnostic methods [8]. However, these methods have several limitations. Moreover, the first three methods require in-depth system knowledge of the system structure and the processes that occur. The others cannot take into account the interaction of the joint or cooperating components.

Machine learning methods have come to occupy an important place in artificial intelligence engineering research. It has become a powerful tool for developing efficient and accurate solutions to many decision-making and optimization problems, based on accumulated data [9,10]. In particular, decision trees have been used to diagnose malfunctions in traditional vehicles [11], for unmanned vehicles researchers have used the support vector machine [12], neural networks [13], the prediction of the unmanned vehicles' behavior has been carried out using the support vector machine [14], decision trees [15], and neural networks [16].

Machine learning does not have a high barrier to entry and the analyst does not need to be a highly skilled expert in automotive design.

We hypothesize that the use of machine learning methods for automotive repair post diagnostics will improve customer service by simultaneously identifying assemblies with a high likelihood of failure.

## 2. Application of Association Rules: A Literature Review

Association rules are one of the machine learning methods, which identify a set of frequent items from a large data set.

Association rules have been used successfully in the past to predict customer purchasing behavior [17,18], analyze bank customer deposit data, analyze school entry scores and three-year average grades [19], predict software failures [20], and identify factors affecting the probability and severity of traffic accidents [21]. The main advantage of association rules is that they can be easily understood and interpreted in programming languages [22].

Because of their ease of interpretation, association rules are widely used as a method of intelligent analysis in a variety of fields. Typical examples include the study of shopping patterns [19], the identification of factors affecting the outcome of accidents [23], disease outbreaks [24], the evaluation of suppliers [25] and software reliability [26], risk prediction for marine logistics services [27], construction of automobile insurance recommendation systems [28,29], and the formation of options combinations chosen when buying cars [30].

It has also been applied in studying the reliability of complex technical systems. Thus, in [31,32], the authors proposed to study the failure correlation coefficients between types of failures and their causes based on failure data of computer numerical control systems with lifetime data using association rules. In [33], rules were used in the detection of anomalous performance of process monitoring systems, such as fluid level monitoring systems; in [34], rules were used for evaluating the reliability of an electric power system. Moreover, ref. [35] was devoted to established fuzzy association rules built for explaining the relationship between the characteristics which can be measured and the failure of an industrial boiler. The authors of [36] used association rules and principal component analysis or support vector data description for finding anomalous performance in an unmanned aerial vehicle system. The authors of [37] proposed, for the first time, to use communication data mining to detect abnormal or malicious vehicle operation.

A structure for mining consistent samples of vehicle maintenance behavior from historical repair records under predefined support and thresholds was proposed in the study by [38]. A rule-based procedure was offered to predict the next maintenance interval and provide necessary information about spare parts. The proposed procedure can be extended for exploring repair actions and conducting root cause analysis to provide maintenance managers with more valuable recommendations on corrective actions to be taken to avoid further breakdown. The timestamp information can also be used to prioritize maintenance activities.

In [39], a correlation analysis was derived from the relationships among the default components, the time of manufacture, the place of operation, and the vehicle type. The researchers used SPSS Clementine with a minimum support value of 0.2% and a minimum confidence level of 20%.

In [40], a power system control device recorded a number of diesel engine thermal parameters, including the master cylinder exhaust temperature, the average exhaust temperature, the scavenging air temperature, the main bearing oil outlet temperature, and the cylinder coolant outlet temperature. Using MATLAB rules, the linking system failures (nozzle, air cooler, oil system, cooling system) and the thermal characteristics connected with engine defaults were created.

Warranty data is a crucial factor in vehicle improvement issues. For example, in [41] the authors used failure statistics accumulated over the last three years on a warranty for heavy excavator equipment and diesel engines to identify correlation patterns between the manufacturing process, the defaults that occurred during the guarantee period, and the subsequent failure series. They used SAS Enterprise Miner 6.2 software to find these patterns.

The algorithm in [42] used the concept of basis sets and database manipulation techniques to establish meaningful associations between vehicle characteristics and failure causes. The association rule represents these relationships, where the first part of the rule contains a set of attributes representing the data about vehicles (production date, repair

date, mileage, transmission, engine type) and the second part of the rule contains the data representing the fault codes associated with failures. Because consistent exploration of warranty data can be very useful for product manufacturers, the authors went further and used association rules in their next study to look for patterns and correlations between follow-up requests to warranty service centers [43].

The future of the automotive industry is linked to the development of smart connected cars, with different reliability requirements than classic cars. As a result, more information in the form of fault codes can be stored in the vehicle's onboard systems. In [44], the authors used the Apriori intelligent association rule analysis algorithm to find and improve the reliability of the fault code rules that result in component failures. Additionally in [45], the authors used association rules to establish the relationship between the diagnostic trouble codes generated during the car's operation and stored in the memory bus, and the codes for repair actions that were taken to eliminate the malfunctions. Previously, they cleared the initial array of training data from anomalous records data on repairs either with a duration that far exceeded the standards, or that provoked repeated visits to service enterprises due to poor-quality troubleshooting, or was associated with unreasonable overspending of spare parts. They implemented their methodology in the prototype of the system, which used a distributed web client-server architecture. We observed the development of this work in [46]. It presented a system of ontology built on the basis of association rules that were applied not only to the database of encoded data on diagnostic faults codes and the date of their fixation, and the work performed, but also to a textual description of the repair, including complaints from the client, regarding used spare parts. The author built three groups of rules: combinations of failed parts, combinations of observed failures symptoms and detected defective parts, and combinations of failure signs for the parts and repair methods applied.

Thus, in many cases, the researchers aimed to find sequential patterns that were designed to identify the relationship between the production conditions and failures and to determine the failures series. However, insufficient consideration has been paid to the diagnostic process, which involves identifying the subcomponents that are likely to fail simultaneously, when examining the interaction of the joint components. Furthermore, when analyzing the works, the researcher only used statistical and analytical platforms to obtain and derive the dependencies, making it difficult to apply the obtained results to real service company situations.

In [47], the authors developed a desktop application in Visual Studio IDE that used, for the analytical process, data on the vehicle name, the vehicle part code, and the faulty part code. These association rules explained some extent the intrinsic correlation between the vehicle assembly components and the faulty parts, but they are not applicable to direct vehicle diagnosis. Therefore, the goal of this study is to develop an artificial intelligence-based methodology that can identify possible failures in the assembled parts, in a timely manner, for direct diagnosis by a repairman.

To achieve this goal, the following issues needed to be addressed:

- Select quality criteria and a rule generation algorithm, determine the requirements for the input data structure and prepare statistical data on the technical defects in the vehicles, including information on the defective parts;
- Design and develop a component reliability analysis web service that allows the generation of association rules interactively and generates a list of parts to be inspected;
- Generate association rules and analyze their quality.

## 3. Materials and Methods

### 3.1. The Essence of the Association Rules Method and Selected Quality Criteria

As applied to our subject area, associative rules are sets of vehicle parts and components that often turn out to be defective together. The simultaneity of the failure is fixed at the time the vehicle enters the service center and is registered with a specific document: a reclamation act if the vehicle is under warranty, and a work order if the vehicle is in the

post-warranty period. The goal of an association rules is to find all frequent sets of parts above a user-defined support threshold and to generate all association rules above different confidence thresholds. The result is a set of related entities called condition M (antecedent) and result N (consequent), written M→N ("M follows N"). Thus, an association rule is expressed in the form "If a condition–default part, then the result–default part is". The quality of each rule is evaluated by its degree of support, confidence, and lift. This choice of criteria was due to the capabilities of the Accord.NET machine learning framework, which was used to develop the module for generating association rules.

The support is defined as the ratio of the number of transactions in which the condition (M) and the result (N) of the rule occur simultaneously to the total number of transactions in the database (W):

$$S(M{\rightarrow}N) = \text{Number } (M \cap N)/W \tag{1}$$

In our case, a transaction is understood as a document containing information about the identified defective parts (reclamation act or work order). The degree of support can take values from 0 to 1.0, which is equivalent to a range of 0–100%. If the support value is S(M→N) = X%, it means that X% of the documents in the database contain a combination of M and N. Thus, the higher the support value, the more often the rule occurs. However, it should be borne in mind that the database usually contains a large number of documents (from several thousand to millions). Therefore, even a small value of support (tenths or hundredths of a percent) is equivalent to a significant value of the absolute number of documents: tens, hundreds, and thousands of examples of the simultaneous appearance of a condition and a result.

The confidence is the ratio of the number of documents containing the condition and the result is the number of documents containing only the condition:

$$C(M{\rightarrow}N) = P\ (M \cap N)/P(M) \tag{2}$$

The confidence level can also take values from 0 to 1.0, which is equivalent to a range of 0–100%. If the support value is C(M→N) = Y%, this means that the document containing the condition M also contains the consequence N in Y% of cases.

The lift is the ratio of the product of the frequency with which a condition and result occur simultaneously and the frequency with which they occur separately:

$$L(M{\rightarrow}N) = P\ (M \cap N)/(P(M){\cdot}P(N)) \tag{3}$$

The lift over 1 is an indication that conditions and results are more likely to occur together in the document than independently. That means, the occurrence of the condition default part in a given claim is more likely to lead to the result default part, and vice versa. If the value of the lift is L = Z, this means that in the document containing the condition M, the result N will occur Z times more often than any other component. A rule with a lift value higher than 1 may be regarded as significant. The lift below 1 is an indication that conditions and results occur individually more often than they occur together in a transaction. In other words, in this case, there is an "anti-rule" where the occurrence of a condition has a negative impact on the occurrence of the outcome. Finally, a high value close to 1 indicates that the conditions and outcomes occur together with the same frequency as they occur separately in a transaction. This implies that the conditions and outcomes do not affect each other's occurrences.

### 3.2. Algorithm for Generating Association Rules

The general approach to generating association rules consists of two enlarged steps. During the first step, sets that occur with a frequency that belongs to the range $(\text{support}_{minimum}, \text{support}_{maximum})$ are formed. During the second step, those sets whose confidence and lift do not belong to the ranges $(\text{confidence}_{minimum}, \text{confidence}_{maximum})$ and $[\text{lift}_{minimum}, \text{lift}_{maximum})$ are deleted.

The ancestor of rule extraction algorithms is the Apriori algorithm [48]. A significant drawback is its low productivity. A complete enumeration of combinations requires a large amount of computational resources and does not allow the quick generation of rules that satisfy the given hyperparameters. Therefore, subsequently researchers have proposed various modifications to this algorithm (AprioriTID and AprioriHybrid) and other methods (FP growth [49], ECLAT [50]) to build efficient processes for extracting frequent sets of elements for association rules.

The work [51], presents the UniqAR algorithm, which allows the generation of unique classification association rules that allow one single class label to be generated and, thus, has 100% confidence. However, as noted in the book [52], "strong rules are not necessarily interesting". After the elimination of rules lower than the given level of support and confidence, we get the so-called strong rules. Indeed, at high levels of support and confidence the resulting rules are generally trivial, and the resulting combinations are well known to the road transport operating specialist. Therefore, the task of selecting the minimum and maximum levels of support, confidence and lift is, in fact, an additional research task in finding the optimal hyperparameters of the association rules algorithm.

Another hyperparameter that affects the speed of work and the composition of the resulting association rules is the power of the rule. This is the number of objects (in our case, parts) that are included in the rule. High values of power, support, confidence, lift, on the one hand, provide the generation of more rules and, on the other hand, require more computing resources, so it is necessary to find a balance between the values of the rule quality indicators and the speed. Moreover, as mentioned above, more rules do not mean they are better.

For our research, we decided to use the Apriori algorithm, which is built into the Accord.NET machine learning framework. On the one hand, it has simplicity, on the other hand, there is no reason to reject it at the beginning of the development of a method for generating combinations of vehicle parts that fail together.

### 3.3. Technique for Detecting Joint Defective Parts

To achieve the research goal, a methodology for detecting and using defective joint components based on association rules was developed. To generate association rules using the Apriori algorithm, it is necessary to prepare a data set for training. This data set should contain records in the form presented in Table 1.

**Table 1.** Structure of the initial data.

| ID | DEFECTED_DETAIL |
|---|---|
| CLAIM _1 | DEFECTED_DETAIL_1 |
| CLAIM _1 | DEFECTED_DETAIL_2 |
| CLAIM _1 | DEFECTED_DETAIL_3 |
| . . . | . . . |
| CLAIM _J | DEFECTED_DETAIL_1 |
| CLAIM _J | DEFECTED_DETAIL_2 |
| CLAIM _J | . . . |
| CLAIM _J | DEFECTED_DETAIL_I |

CLAIM is understood as the number of the document that includes the results of diagnosing a vehicle when it is contacted by a service center (reclamation report or work order). In fact, such a structure of initial data is easy to obtain from the information system database of a vehicle service enterprise, where information about customer requests and repair results is entered.

Further training is carried out, as a result of which an array of rules is obtained. The structure of the rules array with a power equal to two has the form presented in Table 2.

**Table 2.** Structure of the resulting association rules.

| ANTECEDENT | CONSEQUENT | SUPPORT, pcs. | SUPPORT*, % | CONFIDENCE, % | LIFT |
|---|---|---|---|---|---|
| DEFECTED_DETAIL_1 | DEFECTED_DETAIL_2 | $S_1$ | $S^*_1$ | $C_1$ | $L_1$ |
| DEFECTED_DETAIL_2 | DEFECTED_DETAIL_4 | $S_2$ | $S^*_2$ | $C_2$ | $L_2$ |
| DEFECTED_DETAIL_3 | DEFECTED_DETAIL_1 | $S_3$ | $S^*_3$ | $C_3$ | $L_3$ |
| . . . | . . . | . . . | . . . | . . . | . . . |

After the formation of such a rule base, it can be used as a recommendation system. When the client enters, the diagnostician starts the inspection and discovers the defective part. Then the rule base is searched for matches with the ANTECEDENT column and one or more rule consequences that include this part as an antecedent are returned. After evaluating the quality criteria, the service specialist can rank the order for the further checking of parts from this list according to descending quality indicators. Thus, the probability that all the defective parts will be detected increases, and the time of the defective process is also reduced.

## 4. Results

### 4.1. The Structure and Algorithms of the Recommendation Web Service

We have developed an AutoAnalytics web service for facilitating the validation of the proposed methodology. It uses a knowledge base of patterns obtained by applying the association rules method to vehicle failure statistics.

The following components were created for the web service:

- A rule generation component that runs in the background on a schedule;
- A component that implements the proposed recommendation mechanism in the form of a preliminary list of faults.

Association rule generation is computationally intensive and, therefore, runs in the background on a schedule.

To generate association rules, data from tables, such as part numbers, work orders, claim reports, etc., are needed. The CLAIM_NUMBER field is used as the transaction (event) number. The DETAIL field is used as data for analysis (Figure 2). The rules are generated according to the methodology described in Section 3.3.

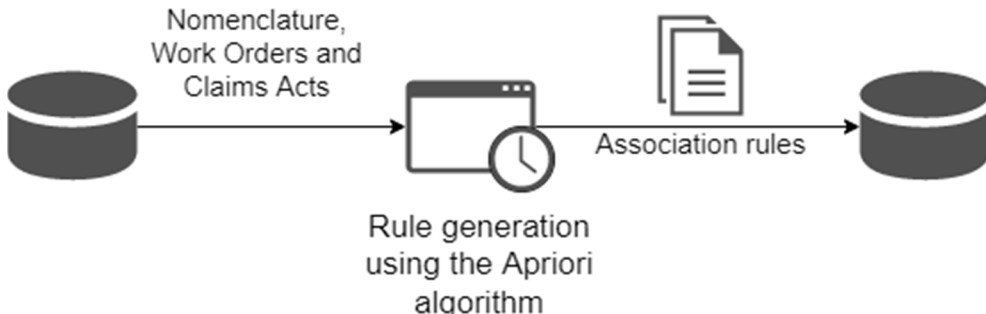

**Figure 2.** Block diagram of the knowledge base generation module.

To extract recommendations, we needed information about the claim and its defective parts, as well as data obtained from the database tables: nomenclature, association Rules (Figure 3). Users send the identifier of the found defective part in a request to the application server, which searches in the records of the association rules table filled in after the work of the previous module, where this part is included as a condition, and gives the client a list of selected parts that are included in the rules as consequences.

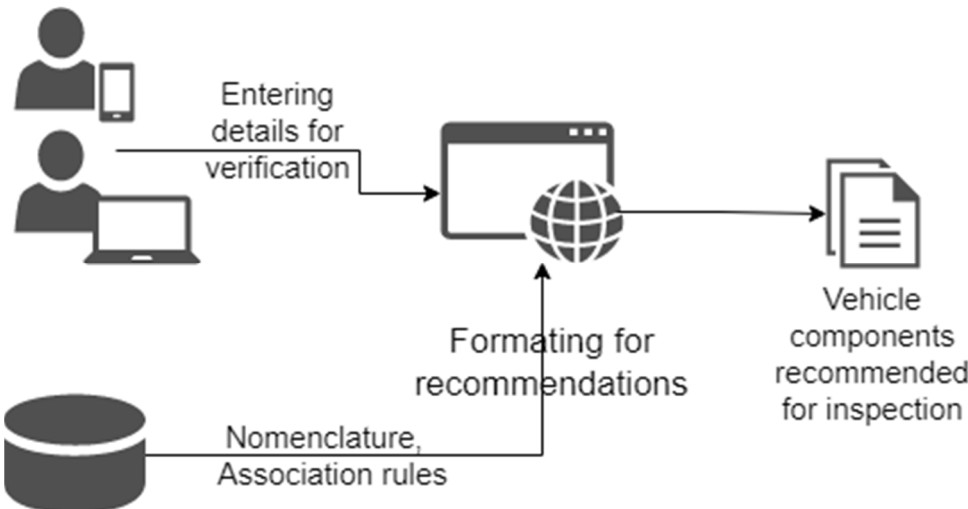

**Figure 3.** Block diagram of the generating recommendations module.

An appropriate database schema was developed (Figure 4) to store the generated association rules, for the purpose of analyzing the dynamics of changes in combinations of defective assemblies, taking into account the type of vehicle and operating zone. For this, the regions_id, model_id fields and the region and model tables are provided.

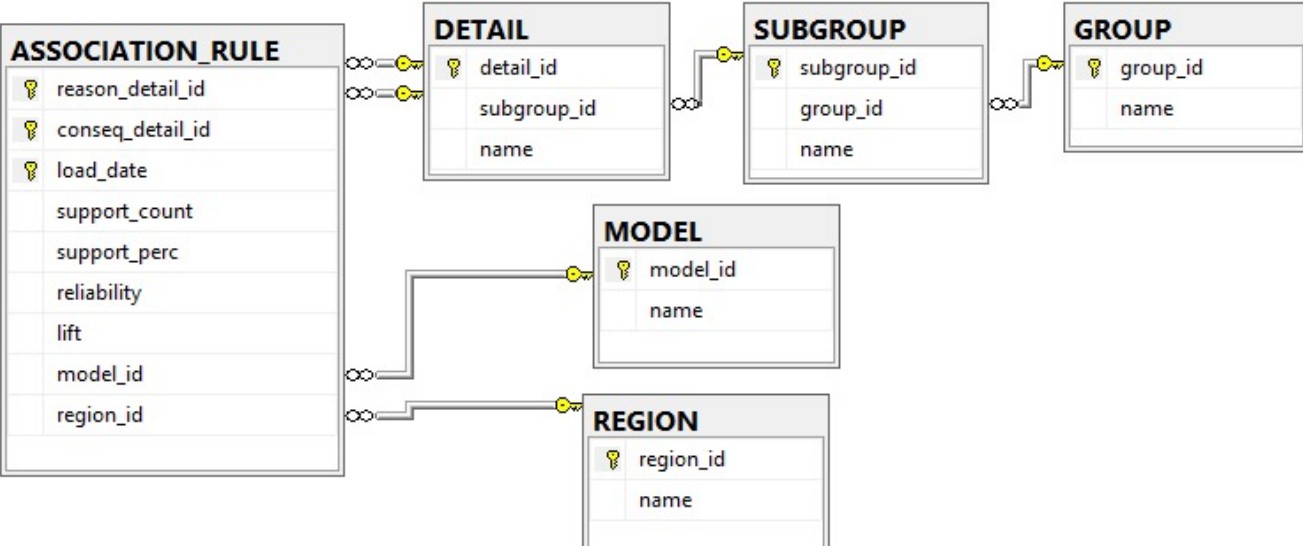

**Figure 4.** Database schema.

The solution was built using a three-tier architecture: database, server, and user interface (Figure 5). As a database management system, we chose PostgreSQL, which is open source and distributed free of charge. We also chose a combination of C# and ASP.NET frameworks because of their suitability for quickly building web applications. Entity Framework Core was used to speed up the development, and simplify and speed up the database operations. When implementing the front-end of the site, HTML and CSS were involved. In addition, JavaScript was used for the site dynamics. To implement association rules, we used the Accord.NET framework, a machine learning platform written entirely in C#. This library allowed us to integrate most of the business logic into the application itself, which made the service architecture easier and simpler.

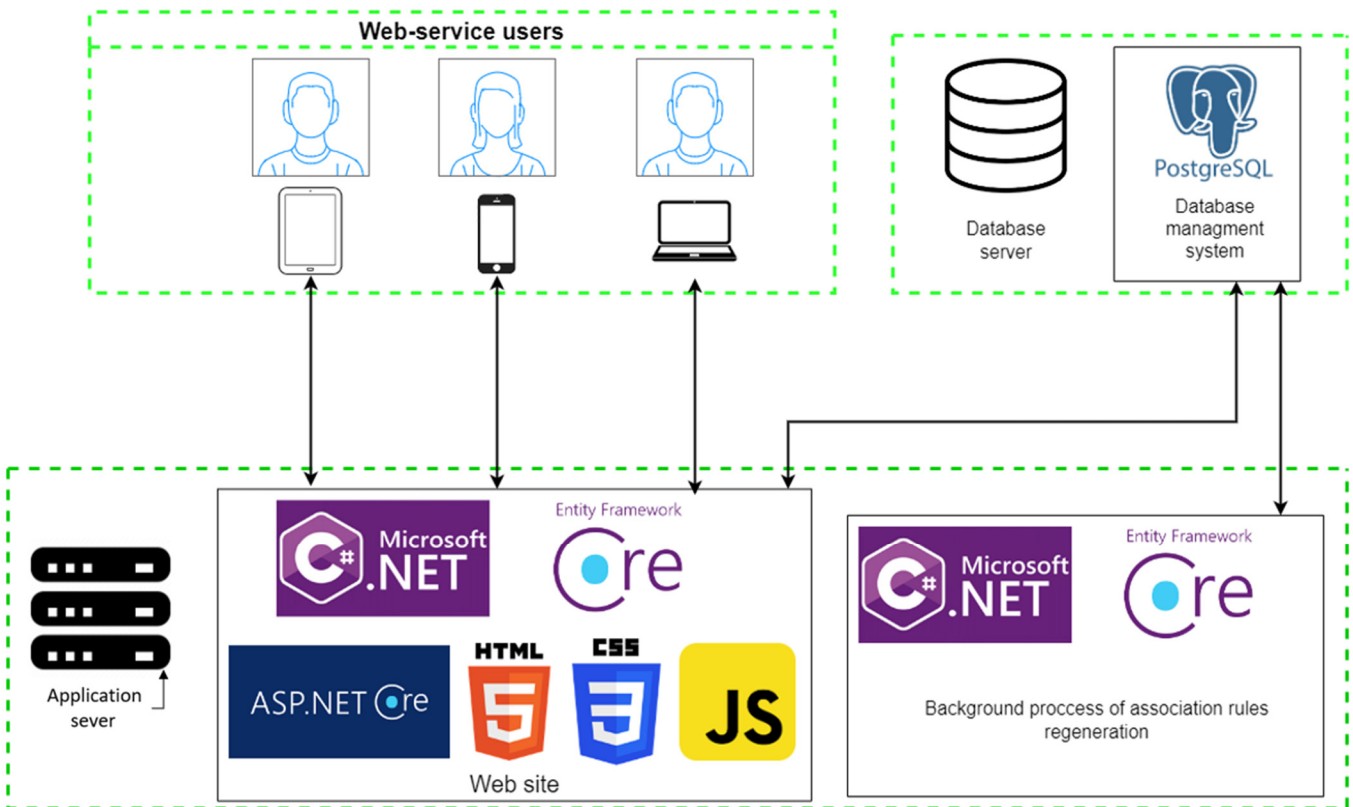

**Figure 5.** Web service architecture and technology stack.

When a fault is detected, the employee enters the corresponding node into the fault list and a list of nodes that are recommended to be checked will be available. For the convenience of the user, a three-stage input of already detected defective parts is implemented: first, the group to which the part belongs should be selected, then the subgroup, and only then the part itself should be selected. For example, if it was found that the pressure sensor is faulty, then, accordingly, we must first select the devices group and the pressure gauges subgroup. The list of parts recommended for inspection is displayed in a similar way. This mechanism is similar to the hierarchical way of organizing the nomenclature directory and is familiar to a warehouse or car service worker. Several assemblies can be added to the set of faults. A list of parts recommended for inspection is sorted according to descending rule validity, where the detected defective parts are listed as "antecedent" (Figure 6). The set of recommended parts is updated as they are added.

By implementing this tool as a web service adapted to mobile platforms, we enable maintenance professionals to use their mobile devices to obtain up-to-date information as they work.

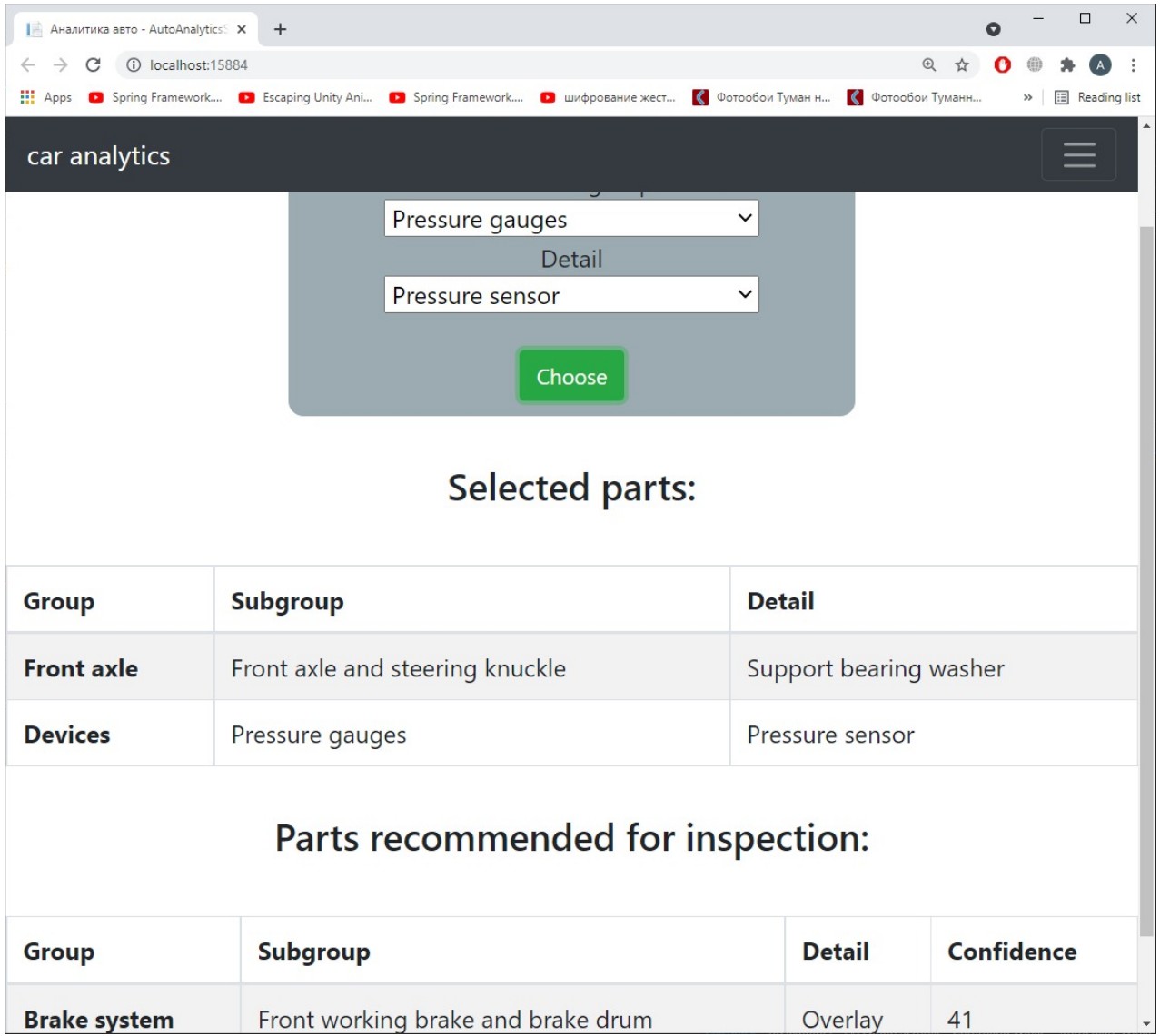

**Figure 6.** AutoAnalytics web service interface.

*4.2. Testing a Web Recommendation Service: Derived Association Rules*

To test this technique, we used real failure statistics for the years 2008–2009, containing 3126 records of failed parts in vehicles that were under warranty service and had a mileage of 1000 km to 10,000 km. This fleet of vehicles operated on the territory of the second category of operating conditions (R1-T4, R2-T1, T2, T3, T4, R3-T1, T2, T3), which means:

Road surfaces:

- R1—improved capital cement concrete, monolithic, reinforced concrete or reinforced prefabricated, asphalt concrete, paving stones and mosaics on a concrete base;
- R2—improved lightweight from crushed stone, gravel and sand treated with binders from cold asphalt concrete;
- R3—transitional crushed stone and gravel.

Terrain type (determined by height above sea level):

- T1—flat, up to 200 m;
- T2—slightly hilly, over 200 to 300 m;
- T3—hilly, over 300 to 1000 m;
- T4—mountainous, over 1000 to 2000 m.

As a result, we obtained a number of non-trivial useful rules. They included rules linking the components of the brake system and transmission system, the electrical system, the electrical system and brake system, and the engine cooling system (Table 3). The confidence level of these rules was high and reached 77%. Although some rules have a 20% confidence level, the high level of lift, which is significantly higher than 1, allowed us to talk about the usefulness of these rules.

**Table 3.** Obtained meaningful association rules.

| Rule Number | Antecedent, Group | Antecedent, Subgroup | Antecedent, Detail | Support, % | Confidence | Lift |
|---|---|---|---|---|---|---|
| | Consequent, Group | Consequent, Subgroup | Consequent, Detail | | | |
| 1 | Brake system | Front working brake and brake drum | Cover | 0.61 | 76.9 | 52.5 |
| | Front axle | Front axle and steering knuckle | Support bearing disc | | | |
| 2 | Front axle | Front axle and steering knuckle | Support bearing disc | 0.61 | 41.7 | 52.5 |
| | Brake system | Front working brake and brake drum | Cover | | | |
| 3 | Brake system | Bypass brake valve | Dual-magister valve | 0.26 | 26 | 9.8 |
| | Devices | Oil pressure gauge | Gauge | | | |
| 4 | Devices | Tyre pressure gauge | Pressure gauge | 0.14 | 21.2 | 8 |
| | Devices | Pressure gauge | Gauge | | | |
| 5 | Electrical equipment | Generator | Relay regulator | 0.10 | 41.7 | 28.8 |
| | Devices | VK403B | Reversing light switch | | | |
| 6 | Cooling system | Fan and its drive | Electromagnetic clutch engagement sensor | 0.08 | 26.6 | 17.2 |
| | Cooling system | Thermostat | Thermostat | | | |

The wear of the brake system components leads to the appearance of vibrations during braking. In this case, there is accelerated wear of the mating elements of the suspension and steering of the vehicle, namely, as follows from rule 1, the support bearing disc. Rule 2 is a mirror of rule 1, which confirms the relationship between the brake system and the running gear.

The analysis of rule 4 did not reveal the relationship between the elements, even indirectly. During troubleshooting, it was revealed that the air pressure drop indicator in the third circuit was on. Inspection of the electrical circuit indicated a failure in the emergency air drop sensor. At the same time, the emergency oil pressure alarm in the engine was constantly on due to a failure in the emergency oil pressure sensor. In both cases, the sensors were found to be replaceable.

According to rule 3, there should have been a connection between the failure of the dual-magister valve, which was detected in the slow release of the parking brake system, and the failure of the emergency oil pressure gauge, which manifested itself in the constant burning of the emergency oil pressure alarm in the engine. However, the analysis of the design does not provide for the identification of patterns between them.

A malfunction in the relay regulator in the generator often contributes to an increase in the voltage of the onboard electrical network, which in turn leads to accelerated wear of its electrical components. This explains the high reliability of rule 5, which in this case was 41.7%, this is the proportion of cases when the relay regulator fails, the reversing light switch is found to be defective.

If the electromagnetic clutch engagement sensor fails, the temperature regime of the engine increases, and the likelihood of it overheating increases. This can explain the accelerated degradation of the thermostat element and rule 6.

Thus, the selected composition of transactions does not allow us to explain the patterns found in all cases. This can be explained either by the absence of such a relationship or by the need to include additional information about both the failures themselves and the vehicles, and the characteristics of the components.

The described method is designed primarily for staff in vehicle repair and maintenance posts. Its use can accelerate the vehicle diagnostic process by providing employees with recommended vehicle parts to check.

It also allows purchasing departments to use well supported rules to select the most promising suppliers and manufacturers, and to develop tactical solutions to eliminate less reliable part suppliers. In addition, the structure of the association rules themselves can be dynamically analyzed to assess the impact of changes made to the design of individual vehicle model parts, as well as the design and manufacture of technology.

## 5. Conclusions

Manufacturers of modern intelligent vehicles are forced to look for new tools and techniques for ensuring the smooth operation of their customers' automobiles. The diagnostic decision support methodology described, based on the relationships obtained, can be used by workers with little experience or qualifications. During the next visit by the vehicle to the service center for maintenance or repair, the employee enters default parts and receives a set of parts made by the application for checking. The aforementioned methodology and the web application developed will improve the quality of and speed up the diagnostics. In addition, designers and engineers can use association rules to analyze possible causes of defective parts based on defects that commonly occur. Automotive manufacturers' purchasing departments can also use such a rules base to select parts suppliers.

Future research will focus on developing the proposed methodology to update the rules base in real-time after each repair and to provide information on the results of defect inspection and the process of repairing the vehicle. Obviously, it needs to investigate the speed of different algorithms for generating association rules.

**Author Contributions:** Conceptualization, P.B.; methodology, P.B.; formal analysis, I.M., A.V. and A.K.; investigation, P.B., I.M., A.V. and A.K.; resources, P.B. and A.V.; software, A.V. and A.K.; writing—original draft preparation, P.B. and A.V.; supervision, P.B.; writing—review and editing, P.B., I.M., A.V. and A.K.; visualization, P.B. and A.V. All authors have read and agreed to the published version of the manuscript.

**Funding:** This research received no external funding.

**Data Availability Statement:** Data acquisition can be discussed with the corresponding author.

**Conflicts of Interest:** The authors declare no conflict of interest.

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
