# Peer review of "The Process of Identifying Automobile Joint Failures during the Operation Phase: Data Analytics Based on Association Rules"

_information, doi:10.3390/info14050257_

Round 1

Reviewer 1 Report

I think this article is more like a software development architecture than a research paper. The proposed approach is biased towards being a concept and does not delve into the causal relationship between fault and features

Author Response

Dear reviewer,

We appreciate your  feedback on our paper. Please see file which explain our responses to the comments, and we highlighted all changes in the manuscript as track changes.

Point 1: I think this article is more like a software development architecture than a research paper. The proposed approach is biased towards being a concept and does not delve into the causal relationship between fault and features.

Response 1: As we indicated in the introduction, classical methods of reliability analysis require deep systemic knowledge about the structure of the system under consideration and the processes occurring in it. Therefore, we proposed to use associative rules - a machine learning method that allows to identify associated defective parts without understanding the cause-and-effect relationship between a malfunction and symptoms. Nevertheless, thank you for the comment, we have added an additional substantiation of the method relevance (L. 39-65) and an analysis of the obtained rules from a technical point of view (L. 448-475).

Best regard,

Authors

Reviewer 2 Report

In the work, the authors proposed an approach based on association rules, a machine learning technique, to simplify the defect detection process. To facilitate its use in a real service company environment, we have developed a web-service that allows a repairman to simultaneously identify nodes with a high probability of failure. The structure and working principles of the developed web-service are described, as well as the procedure for its application, which resulted in the discovery of several useful non-trivial rules. Several rules resulting from the use of this interactive tool are presented, which allow repairers to quickly  and easily detect possible defects in relevant components during the diagnostic process. These rules  are also highly supported and can be used by procurement departments to make tactical decisions  when selecting the most promising suppliers and manufacturers. The methodology developed allows the evaluation of the effectiveness of changes in design, technology for the manufacture and operation of individual vehicle components, analyzing the change in the composition of combinations of parts over time. Generally, this is a good work. It can be accepted if the authors can consider the following issues: 1. It seems that the topic is not suitable.  Is the proposed method applicable when the  automobiles are produced or  when the  automobiles are used? 2. What is the main failure mode? 3. More words are welcome for all the figures. 4. More related works on the faults identifications are welcome to enrich the review such as Fault diagnosis of an autonomous vehicle with an improved SVM algorithm subject to unbalanced datasets;An Improved Learning-Based LSTM Approach for Online Lane Change Intention Prediction Subject to Imbalanced Data. 5. The language should be improved significantly.

Author Response

Dear reviewer,

We appreciate your  feedback on our paper. Please see file which explain our responses to the comments, and we highlighted all changes in the manuscript as track changes.

Response to Reviewer 2 Comments

Point 1: It seems that the topic is not suitable.  Is the proposed method applicable when the  automobiles are produced or  when the  automobiles are used?

Response 1: We have reformulated the topic: “The process of identifying automobiles joint failures during the operation phase: data analytics based on association rules”.

Point 2: What is the main failure mode? 

Response 2: We have added this information (L. 426-440).

Point 3: More words are welcome for all the figures. 

Response 3: We have added.

Point 4: More related works on the faults identifications are welcome to enrich the review such as Fault diagnosis of an autonomous vehicle with an improved SVM algorithm subject to unbalanced datasets;An Improved Learning-Based LSTM Approach for Online Lane Change Intention Prediction Subject to Imbalanced Data.

Response 4: We have added (L. 77-80).

Point 5: The language should be improved significantly.

Response 5: We have tried to improve.

Best regard,

Authors

Reviewer 3 Report

A good paper for begin with; i hope the continuation with a greater number of results.

Results showed are howewer sufficient for validation.

Just a little advert: at Table 3 must to be writed "gauge" instead "gage".

Author Response

Dear reviewer,

We appreciate your  feedback on our paper. Thank your for your high score. We have fixed the spelling errors, and we have highlighted all changes in the manuscript as track changes.

Response to Reviewer 3 Comments

Point 1: Just a little advert: at Table 3 must to be writed "gauge" instead "gage".

Response 1: We have corrected it. Thank you.

Best regard,

Authors

Round 2

Reviewer 1 Report

It can be accpeted.